# Metabolomics and Transcriptomics Provide Insights into Lipid Biosynthesis in the Embryos of Walnut (*Juglans regia* L.)

**DOI:** 10.3390/plants12030538

**Published:** 2023-01-24

**Authors:** Manman Liang, Xuemei Zhang, Qinglong Dong, Han Li, Suping Guo, Haoan Luan, Peng Jia, Minsheng Yang, Guohui Qi

**Affiliations:** 1College of Forestry, Hebei Agricultural University, Baoding 071001, China; 2Technology Innovation Center of Hebei Province, Xingtai 054000, China; 3Institute of Walnut Industry Technology of Hebei Province (Xingtai), Lincheng 054300, China

**Keywords:** oil content, lipid biosynthesis, *SAD*, FAD2, regulatory network

## Abstract

Walnut (*Juglans regia* L.) is an important woody oilseed tree species due to its commercial value. However, the regulation mechanism of walnut oil accumulation is still poorly understood, which restricted the breeding and genetic improvement of high-quality oil-bearing walnuts. In order to explore the metabolic mechanism that regulates the synthesis of walnut oil, we used transcriptome sequencing technology and metabolome technology to comprehensively analyze the key genes and metabolites involved in oil synthesis of the walnut embryo at 60, 90, and 120 days after pollination (DAP). The results showed that the oil and protein contents increased gradually during fruit development, comprising 69.61% and 18.32% of the fruit, respectively, during ripening. Conversely, the contents of soluble sugar and starch decreased gradually during fruit development, comprising 2.14% and 0.84%, respectively, during ripening. Transcriptome sequencing generated 40,631 unigenes across 9 cDNA libraries. We identified 51 and 25 candidate unigenes related to the biosynthesis of fatty acid and the biosynthesis of triacylglycerol (TAG), respectively. The expression levels of the genes encoding Acetyl-CoA carboxylase (ACCase), long-chain acyl-CoA synthetases (LACS), 3-oxoacyl-ACP synthase II (KASII), and glycerol-3-phosphate acyl transfer (GPAT) were upregulated at 60 DAP relative to the levels at 90 and 120 DAP, while the stearoyl-ACP-desaturase (*SAD*) and fatty acid desaturase 2 (*FAD2*) genes were highly abundantly expressed during all walnut developmental periods. We found that ABSCISIC ACID INSENSEITIVE3 (ABI3), WRINKLEDl (WRI1), LEAFY COTYLEDON1 (LEC1), and FUSCA3 (FUS3) may be key transcription factors involved in lipid synthesis. Additionally, the metabolomics analysis detected 706 metabolites derived from 18 samples, among which, 4 are implicated in the TAG synthesis, 2 in the glycolysis pathway, and 5 in the tricarboxylic acid cycle (TCA cycle) pathway. The combined analysis of the related genes and metabolites in TAG synthesis showed that phospholipid:diacylglycerol acyltransferase (*PDAT*) genes were highly abundantly expressed across walnut fruit developmental periods, and their downstream metabolite TAG gradually accumulated with the progression of fruit development. The *FAD2* gene showed consistently higher expression during fruit development, and its downstream metabolites 18:2-PC and 18:3-PC gradually accumulated. The *ACCase*, *LACS*, *SAD*, *FAD2*, and *PDAT* genes may be crucial genes required for walnut oil synthesis. Our data will enrich public databases and provide new insights into functional genes related to lipid metabolism in walnut.

## 1. Introduction

Walnut (*Juglans regia* L.) belongs to the Juglandaceae family, which is an important and widely cultivated woody oilseed tree species [1]. As a functional oil, walnut oil exerts various effects, including an overall health-promoting effect on the human body [2,3,4]. Its many health benefits include improving cognitive function and intelligence, lowering blood lipids, lowering blood glucose, anti-oxidation, and anti-cancer effect [5,6]. In recent years, the high content of oil and polyunsaturated fatty acids in walnut fruit has attracted research interest. The oil stored in plant seeds mainly exists in the form of TAG [7]. The synthesis of vegetable oil mainly involves the synthesis of fatty acids in the plastids and the synthesis of TAGs in the endoplasmic reticulum. Shi et al. [8] have shown that there were five kinds of main fatty acids in the mature walnut seeds: palmitic acid (16:0), stearic acid (18:0), oleic acid (18:1), linoleic acid (18:2), and linolenic acid (18:3). During the period of oil rapid accumulation, the content of saturated fatty acids decreased rapidly, while the content of unsaturated fatty acids increased rapidly [9].

The first step of the process is the synthesis of fatty acids in the plastid, as follows. ACCase catalyzes acetyl-CoA to produce malonyl-CoA. Malonyl-CoA and acetyl-CoA are then used as substrates to synthesize acyl carbon chains under the catalysis of the fatty acid synthetase complex (FAS) [10]. Then, saturated fatty acids (16:0-ACP and 18:0-ACP) containing ACP are produced, and fatty acyl-ACP thioesterase (FAT) releases the fatty acids. The free fatty acids are subsequently activated by LACS to form acyl-CoA, which is then transported to the endoplasmic reticulum [11,12]. The second step is the synthesis of TAGs in the endoplasmic reticulum. Lysophosphatidic acid acyltransferase (LPAAT), GPAT, and diacylglycerol acyltransferase (DGAT) transfer the fatty acids with CoA to glycerol, and form TAG [13]. Ultimately, TAG combines with oil body proteins to form oil bodies within the cell [14].

Previous studies have shown that the genes LEAFY COTYLEDON1,2 (LEC1, LEC2), FUS3, ABI3, and WRI1 encode key transcription factors involved in oil accumulation [15]. The mutation, overexpression, or ectopic expression of these transcription factors consequently affect seed development and oil accumulation [16]. LEC1 can regulate the expression levels of *FUS3* and *ABI3* to control seed storage protein accumulation [17]. LEC2 can affect the synthesis of fatty acids and TAG by regulating genes such as fatty acid desaturase *(FAD)*, *DGAT*, *GPAT*, *LPAAT*, and phosphatidate phosphatase (*PAP*) [18]. LEC2 is considered a key upstream transcription factor that can directly regulate the expression of *WRI1* [19]. The overexpression of *WRI1* can increase vegetable oil content [20]. It is involved in the transcriptional regulation of carbon source utilization in fatty acid synthesis in seeds, and its regulatory targets are pyruvate dehydrogenase (*PDH*), 3-oxoacyl-ACP synthase I (*KASI*), and *ACCase* [21].

Zhao et al. [22] constructed miRNA-mRNA regulatory modules involved in walnut oil accumulation. A total of 28 genes related to walnut oil biosynthesis were screened out, corresponding to 17 differential expression miRNAs. Huang et al. [23] and Shi et al. [8] have sequenced transcriptomes during walnut seed kernel development, which increases the available transcriptome data for walnut seed kernels’ lipid metabolism. Recently, walnut genome assembly at the chromosome level was published, which will be more conducive to walnut transcriptome research [24]. Previous studies have shown that the transcriptome has been frequently used to identify key genes involved in certain metabolite biosynthesis pathways in plants due to its effectiveness, accuracy, and reliability [25,26]. The integration and expression of exogenous genes affect the metabolic profile of plants to a certain extent [27,28]. Due to the complexity and integrity of biological processes, single-omics data cannot adequately analyze the macroscopic development processes of biological systems. The analysis of multi-omics mutual verification can be used to quickly identify functional genes related to metabolic processes. In recent years, metabolomics integrated with transcriptomics has been widely used to investigate the biosynthesis of metabolites to reveal the biosynthesis pathways of metabolites in plants [29,30,31].

However, until now, little is known about metabolites in walnut embryo development. To systematically understand the regulatory network between metabolites and genes involved in lipid biosynthesis, using transcriptomics integrated with metabolomics during the accumulation period of walnut oil. “Lvling” walnut fruits at three stages (60, 90, and 120 DAP) were selected to perform analysis. Our results will enrich the walnut omics database and provide useful information for plant oil synthesis.

## 2. Results

### 2.1. Analysis of Walnut Fruit Morphological Characteristics and Oil Content

The morphological characteristics of the walnut fruits are shown in Figure 1A. We observed that the fruit diameter gradually increased at 50–70 DAP, thereafter remaining largely unchanged at 70–140 DAP. The embryo was just visible to the naked eye at 50 DAP. The endocarp had not yet formed at 50 DAP. The embryo began to solidify and gradually mature at 60–140 DAP. The dynamic changes in the oil and protein content during the two walnut oil accumulation periods (60–90 and 90–140 DAP) are shown in Figure 1B. The oil content increased significantly from 2.61% to 69.61% during the walnut oil accumulation periods. The oil content increased rapidly at 60–90 DAP and then increased slightly at 90–140 DAP. The protein content increased significantly from 10.25% to 21.12% at 50–90 DAP, while the protein content decreased gradually from 21.12% to 18.32% at 90–140 DAP. The dynamic changes in the soluble sugar and starch content during the walnut oil accumulation periods are shown in Figure 1C. During the period of rapid oil accumulation (60–90 DAP), the soluble sugar and starch content decreased rapidly, then reached a plateau of 90–140 DAP. The contents of soluble sugar and starch reached their highest levels at 50 DAP (18.62% and 13.64%, respectively) and lowest levels at 140 DAP (2.14% and 0.84%, respectively).

### 2.2. Analysis of the Principal Component Analysis (PCA)

According to the embryo morphology development and the oil content, walnut fruits at 60, 90, and 120 DAP were chosen for transcriptomics (each stage had three biological replicates) and metabolomics (each stage had six biological replicates) analysis, designated as L1, L2, and L3, respectively. PCA was conducted to reflect the characteristics of multi-dimensional omics data through several principal components. As shown in Figure 2A, the PCA of all 9 samples was conducted using RNA-seq FPKM. The PC1 and PC2 explained 98.09% and 1.47% of the variation, respectively. Furthermore, L1 was clearly distinguishable from the other two groups (L2 and L3) in the PC1 score plots, while L3 was clearly distinguishable from the other two groups (L1 and L2) in the PC2 score plots. As shown in Figure 2B, the PCA of all 18 samples was conducted based on the peak area. The PC1 and PC2 explained 70.89% and 20.4% of the variation, respectively. Furthermore, L1 was clearly distinguishable from the other two groups (L2 and L3) in PC1 and PC2 score plots. The repeated samples were compactly distributed, thus indicating that there was a high consistency between the biological replicates and that the experiment was reproducible and reliable.

### 2.3. Analysis of Differentially Expressed Genes (DEGs) across Different Developmental Periods

To understand the potential molecular synthesis mechanisms involved in walnut oil fruit development and ripening, nine libraries from walnut embryos were sequenced. A total of 444,613,638 raw reads were obtained and 61.52 G clean bases were obtained after quality control (Appendix A). All of the clean reads were mapped to the walnut genome sequence, and the proportion of the total map rate ranged from 94.77% to 96.87% (Appendix A).

A cluster analysis of the DEGs based on FPKM is presented in Figure 3A. Nine samples were divided into two main groups: samples from the first stage (60 DAP) constituted one group; samples from the last stage (120 DAP) formed the second group. These two groups correspond to the beginning of oil accumulation and the maturity of oil accumulation. The result showed that there were more highly expressed genes at the beginning of oil accumulation. The statistical results of the DEGs are shown in Figure 3B. The number of DEGs in the L2 vs. L1, L3 vs. L1, and L3 vs. L2 comparison groups were 9978, 9358, and 3932, respectively; the corresponding number of those upregulated were 352, 965, and 3050, while those downregulated were 9626, 8393, and 882, respectively. The Venn diagram in Figure 3C shows the overlap of DEGs among the comparison groups. There were 897 common DEGs among all comparisons.

### 2.4. Analysis of Differentially Expressed Metabolites (DEMs) across Different Developmental Periods

Multiomics analysis can more clearly identify the genes regulating walnut oil metabolism, it was used in the present study. The samples collected from the walnut embryos during oil accumulation periods (60, 90, and 120 DAP) were subjected to LC-MS analysis. The total ion chromatogram (TIC) of all the samples showed high stability, large peak capacity, and good retention time (Appendix A).

The cluster analysis of DEMs based on the peak area is shown in Figure 4A. All biological replicates clustered together, indicating the high reliability of the resulting metabolomic data. Among them, most metabolites showed high levels of accumulation in the early stage of fruit development (60 DAP), while some exhibited high levels of accumulation at the maturity stage (120 DAP). This indicates that the accumulation of these metabolites was related to the development and maturation of walnut fruits. The statistical results of DEMs are shown in Figure 4B. The number of DEMs in the L2 vs. L1, L3 vs. L1, and L3 vs. L2 comparison groups were 316, 319, and 225, respectively; the corresponding number of those upregulated were 156, 131, and 74, while those downregulated were 160, 188, and 151, respectively. The Venn diagram analysis of the three comparison groups is shown in Figure 4C. There were 65 common DEMs in all comparisons. 

### 2.5. Analysis of Transcription Factors

Transcription factors (TFs) are important proteins that act as upstream regulators of genes and play key roles in the regulation of plant development. A total of 384 differentially expressed TFs belonging to 27 families were identified across the walnut fruit development stages (Figure 5A, Appendix A). The classified results indicate that most of these TFs belonged to the bHLH (83 members), ERF (66 members), G2-like (30 members), and bZIP (28 members) families. As shown in Figure 5B, the TF family members were either significantly upregulated or downregulated to varying degrees.

### 2.6. Integrated Analysis of the Transcriptomic and Metabolomic Data Pertaining to Lipid Biosynthesis across Walnut Fruit Developmental Periods

To understand the differences in the oil accumulation across fruit developmental periods, transcriptomic and metabolomic data were integrated for analysis (Figure 6, Appendix A). ACCase consists of three subunits, including biotin carboxylase (BC), biotin carboxyl carrier protein (BCCP), and carboxyl transferase subunit alpha (CTα) [13]. Our analysis showed that the expression levels of the genes encoding three subunits (BC, BCCP, and CTα) were upregulated at 60 DAP relative to their levels at 90 and 120 DAP. This may partially explain why oil content increased from 12.38% (60 DAP) to 69.14% (120 DAP). There were three kinds of 3-oxoacyl-ACP synthase (KAS), including KASI, KASII, and KASIII, which can each catalyze the synthesis of different carbon chains in plastids. KASIII catalyzes acetyl-CoA and malonyl-ACP to form 4:0-ACP; KASI catalyzes the synthesis from 4:0-ACP to 16:0-ACP; KASII catalyzes the synthesis from 16:0-ACP to 18:0-ACP [32]. Our analysis revealed that the expression levels of the genes encoding KASI, KASII, and KASIII were upregulated at 60 DAP relative to their levels at 90 and 120 DAP. However, one of the five KASII-encoding genes was more highly expressed at 90 DAP than at 60 and 120 DAP. These expression patterns provided an explanation for the continuous and rapid oil accumulation during early fruit development (60 DAP).

SAD is a key enzyme that catalyzes the transformation of saturated fatty acids to unsaturated fatty acids and can catalyze the transformation of 18:0-ACP to 18:1-ACP [33]. Three *SAD* genes were detected during walnut oil accumulation, one of which had the highest expression level at 90 DAP (with an FPKM value of 580.05). The highly expressed *SAD* may rapidly catalyzed the synthesis of unsaturated fatty acids. FAT consists of two types, including FATA (fatty acyl-ACP thioesterase A) and FATB (fatty acyl-ACP thioesterase B) [23]. FAT has a large influence on the composition of fatty acids, the specificity of their carbon chain lengths, and their activity [34]. FATA has the highest activity for C18:1-ACP, and FATB mainly hydrolyzes the thiolipid bond of the saturated fatty ACP (C16:0-ACP and C18:0-ACP) [35], thereby releasing free fatty acids and ACP. The expression levels of the genes encoding FATA and FATB were upregulated at 60 DAP relative to their levels at 90 and 120 DAP. *FATA* was expressed at substantially higher levels than *FATB*, which is expected to have been beneficial to the accumulation of unsaturated fatty acids. LACS plays an important role in lipid synthesis. In this study, the *LACS* genes were found to be downregulated during walnut oil accumulation periods and had the highest expression levels at 60 DAP.

The FAD is localized to the endoplasmic reticulum, and can convert monounsaturated fatty acids to polyunsaturated fatty acids [36]. Two *FAD2* were detected in this study; one was downregulated during walnut oil accumulation, while the other was characterized by consistently higher expression levels during walnut oil accumulation; it had the highest expression level at 90 DAP (with an FPKM value of 1877). The contents of the downstream metabolites 18:2-PC and 18:3-PC were more highly accumulated at 120 DAP compared to 60 and 90 DAP. This may be related to the persistently high expression of *FAD2* gene. Previous studies showed that the high expression of *PDAT* was beneficial to the accumulation of polyunsaturated fatty acids [37,38]. Seven PDAT-encoding genes and one DGAT1-encoding gene were detected during walnut oil accumulation. Among them, six *PDAT* genes showed a downregulated expression pattern during walnut oil accumulation, while notably, the other *PDAT* gene showed the highest expression level at 90 DAP. The *DGAT1* gene exhibited a low expression level at all developmental stages. The content of the downstream metabolite TAG increased gradually over with the progression of fruit development. Most of the genes showed higher expression levels in the early stage of oil accumulation and lower expression levels in the middle and late stages of oil accumulation, which corresponded to the oil accumulation rate. These crucial genes have similar expression profiles and may play a critical joint role in lipid formation.

### 2.7. Identification of TFs Related to Oil Biosynthesis 

LEC1, LEC2, ABI3, and FUS3 play key roles in seed development and lipid synthesis [23]. Figure 7 shows that the expression level of *LEC1*, *FUS3*, *ABI3*, and *WRI1* were more highly expressed at 60 DAP compared to 90 and 120 DAP. It is worth noting that the FPKM values of *LEC2* were lower than 0.37 across the three stages. The average FPKM value of *LEC1* was 439.59 at 60 DAP, which was much higher than that of the other transcription factors. We speculate that LEC2 plays a minor role in walnut oil accumulation, while LEC1 plays a more critical role.

The correlation analysis between lipid synthesis-related genes and key transcription factors is shown in Figure 8. *WRI1* expression was found to be positively correlated with the expression of the genes pyruvate dehydrogenase E1 component beta subunit (*PDH-E1β*), enoyl-ACP reductase (*ENR*), *KASI*, *KASII*, *KASIII*, malonyl CoA-acyl carrier protein transacylase (*MCAT*), 3-oxoacyl-ACP reductase (*KAR*), *PDAT*, *FATB*, *LACS*, *FAD2*, *LPAAT*, and *SAD*. *ABI3* expression was positively correlated with that of 3-hydroxyacyl-ACP dehydratase (*HAD*), *LACS*, *FATA*, *GPAT*, *PDAT*, pyruvate dehydrogenase E1 component alpha subunit (*PDH-E1α*), *KAR*, *KASIII*, and *DGAT1*. *FUS3* expression was positively correlated with that of *PDH-E1α*, *KAR*, *GPAT*, *LPAAT*, *ACC/BCCP*, *SAD*, *FATA*, *LACS*, and *PDAT*. *LEC1* expression was positively correlated with that of *PDH-E1β*, *ENR*, *KASI*, *KASII*, *KASIII*, *FATB*, *SAD*, *FATA*, *ACC/BC*, *MCAT*, and *FAD2*. 

### 2.8. Integrated Analysis of the Transcriptome and Metabolome of the Glycolysis and TCA Cycle Biosynthesis Pathways across Different Developmental Periods

#### 2.8.1. Analysis of the Glycolysis Pathway

Carbohydrate anabolism comprises much of the basic metabolism of plants, providing carbon sources and energy inputs for lipid synthesis. Pyruvate produced by glycolysis produces acetyl-CoA under the action of PDH, which is the raw material for fatty acid synthesis [39]. To understand the differences in the glycolysis pathway across fruit developmental periods, transcriptomic and metabolomic data were integrated for analysis (Figure 9, Appendix A). The expression levels of the genes encoding pyruvate decarboxylase (PDC), alcohol dehydrogenase (ADH), and aldehyde dehydrogenase (ALDH) were found to be downregulated during walnut oil accumulation. The content of the downstream metabolite acetaldehyde decreased with progressing fruit development, which suggests that these genes may directly affect acetaldehyde content. As the expression of the gene encoding pyruvate kinase (PK) decreased, so too did the content of its downstream metabolite, pyruvic acid. The upstream gene expression profile was consistent with the changes observed in the downstream metabolite content. 

#### 2.8.2. Analysis of the TCA Cycle Pathway

The TCA cycle represents the center of sugar and lipid metabolism. To understand the differences in the TCA cycle across fruit developmental periods, transcriptomic and metabolomic data were integrated for analysis (Figure 10, Appendix A). PDH catalyzes the conversion of pyruvate to acetyl-CoA and is composed of four subunits (E1α, E1β, E2, and E3) [39]. In this study, *PDH-E1α* and *PDH-E1β* were detected, and the expression levels of these genes were increasingly downregulated with continued fruit development and ripening. The expression levels of the genes coding ATP-citrate synthase (ACLY), aconitate hydratase (ACO), malate dehydrogenase (MDH), fumarate hydratase (FH), succinyl-CoA synthetase alpha subunit (LSC1), succinyl-CoA synthetase beta subunit (LSC2), succinate dehydrogenase (ubiquinone) iron-sulfur subunit (SDHB), succinate dehydrogenase (ubiquinone) flavoprotein subunit (SDHA), isocitrate dehydrogenase (NADP) (IDH1), and isocitrate dehydrogenase (NAD+) (IDH3) decreased gradually, and the contents of the downstream metabolites citric acid, s-malate, succinic acid, and oxoglutaric acid were found to decrease gradually with the progression of fruit development. The changes in the upstream gene expression and downstream metabolite content were consistent.

### 2.9. RT-qPCR Validation of DEGs in Transcriptomic Data

In order to validate the transcriptomic data, 12 DEGs were selected for expression analysis by using qRT-PCR. Their expression levels were consistent between the qRT-PCR and RNA-seq data, thereby validating the reliability of the transcriptomic data (Figure 11).

## 3. Discussion

In this study, as the walnut fruits developed and matured, their oil and protein contents increased rapidly at 50–90 DAP and remained relatively stable at 90–140 DAP. On the contrary, their soluble sugar content decreased rapidly at 50–90 DAP and remained relatively stable at 90–140 DAP. This may be attributable to the intermediate products of sugar metabolism providing the necessary raw materials and energy required for lipid synthesis, whereby the sugar content was gradually consumed as the oil content gradually accumulated. According to the changes in the physiological indicators during fruit development, we found that 50–90 DAP was the most critical period for oil synthesis. In this study, enzyme genes and metabolites involved in oil synthesis were detected by transcriptomic and untargeted metabolomic analysis during walnut oil accumulation. 

In this study, 9 cDNA libraries for transcriptome sequencing of walnut embryos at three developmental stages were constructed, and more than 94% of clean reads in each library mapped to the walnut reference genome. Transcriptome analysis showed that the oil accumulation rate in walnut embryos was related to the specific expression of lipid biosynthesis genes. Previous studies on “Linzaoxiang” [23] and “Qingxiang” [40] walnut embryos indicated that *KASIII*, *KASI*, *FATA*, *FATB*, *SAD*, and *FAD2* may play a critical role in lipid synthesis. These genes showed similar expression trends in different walnut varieties. In this study, by analyzing the expression of key enzyme genes involved in oil biosynthesis, it was found that the genes *LACS* and *ACCase* had higher transcript abundance in the oil accumulation periods, and so may play a role in accelerating the biosynthesis of fatty acids. The extension of fatty acid carbon chains, pre-desaturation of fatty acids, transport of fatty acids, and generation of other lipid derivatives from fatty acids all depend on LACS [41]. In this study, the persistently high expression of *FAD2* could accelerate the accumulation of 18:2-PC and 18:3-PC metabolites. We speculated that the *FAD2* gene could also accelerate the accumulation of oil by the TAG synthesis pathway related to the *PDAT* gene.

DGAT and PDAT represent the last step in the catalytic synthesis of TAG, which may be the key step in the synthesis of lipids [42]. The expression level of *PDAT* in walnut embryos is much higher than that of *DGAT1* and *DGAT2* [23]. In *Torreya grandis* kernels, *PDAT* showed a higher correlation with oil content than *DGAT* [43]. The loss of DGAT1 protein function can lead to a change in the composition of fatty acids [44], while its overexpression in *Arabidopsis* has been shown to cause seeds to become larger and have an increased oil content [45]. PDAT is another pathway involved in TAG synthesis, which codes for a key enzyme in the TAG synthesis pathway independent of acyl-CoA [46]. Further, in this study, the expression level of *PDAT* was much higher than that of *DGAT1* during walnut oil accumulation. The high expression of *PDAT* may have more influence on the accumulation of walnut oil.

In addition, GPAT and LPAAT are also key rate-limiting enzyme genes in the TAG synthesis, and previous studies have shown that the overexpression of the yeast *LPAAT* gene in *Arabidopsis thaliana* can promote the accumulation of oil [47]. In this study, the expression pattern of these two genes was downregulated with fruit development and ripening. Therefore, the *GPAT* and *LPAAT* genes might increase oil content by increasing the flow rate of intermediates in the Kennedy pathway. WRI1 and FUS3 may be crucial in the regulation of lipid biosynthesis in *Torreya grandis* kernel [43]. The expression pattern of *WRI1* aligned with the accumulation of walnut oil [23]. The expression patterns of TFs; such as *LEC1*, *FUS3*, *ABI3*, and *WRI1*; were consistent with the pattern of walnut oil accumulation, thus, these TFs may play a critical role in walnut lipid synthesis. In this study, these genes (*Accase*, *FATA*, *TATB*, *KASI*, *KASIII*, *GPAT*, *PDAT*, *LEC1*, and *WRI1*) showed similar expression trends during walnut oil accumulation, and these genes may be related to the development of the walnut embryo.

Sugar metabolism is highly important for the synthesis of plant lipids, and a series of intermediate products of sugar metabolism can provide the necessary raw materials and energy for lipid synthesis during fruit development [48]. The expression of key genes in glucose metabolism may affect fruit oil accumulation [49]. The genes *ENO* and *PK* were highly abundantly expressed and are involved in the glycolytic pathway. The upstream *ENO* and *PK* expression profiles were consistent with the changes observed in the content of downstream metabolite pyruvic acid. In the TCA cycle pathway, the expression of the genes *PDH* were the highest at 60 DAP; their products function to rapidly convert pyruvate to acetyl-CoA and provide sufficient raw materials for fatty acid synthesis. The molecular mechanism of walnut lipid synthesis was systematically studied by analyzing the regulatory relationship between key gene expression profiles and metabolite content during the walnut oil accumulation stage. The qPCR analysis confirmed that the expression data obtained by RNA-Seq was reliable. The findings provide a theoretical reference for the improved breeding of high-lipid varieties of walnut or other means of walnut lipid synthesis. Nevertheless, the regulating mechanism of oil accumulation in developing embryos is still unclear. Further studies on their functions in FA biosynthesis and oil accumulation are required.

## 4. Materials and Methods

### 4.1. Plant Materials

Walnut fruits were collected from a 19-year-old walnut cultivar “Lvling” trees from Hebei Lvling Fruit Industry Co., Ltd. (114°30′-114°33′ E, 37°29′-37°32′ N, elevation 90–135 m), which resides 6 km north of Lincheng County, Hebei Province, China. Walnut fruits with a similar growth state that was free of pests and diseases were selected for sampling. The trees were artificially pollinated on April 10 (0 DAP). Fruit samples were collected during the embryo development stage from June 1 to August 29, 2020. The fruits were harvested at 50 DAP (i.e., when the embryos were only just visible) and then every 10 days until 140 DAP (i.e., at embryo maturity). Twenty mixed fruits obtained from three trees were regarded as one biological replicate. Three biological replicates were collected for transcriptome and physiological indices analysis, and six biological replicates were collected for untargeted metabolome analysis. Walnut fruits at 50, 60, 70, 80, 90, 100, 110, 120, 130, and 140 DAP were selected for the analysis of physiological indices. Walnut fruits at 60, 90, and 120 DAP were selected as transcriptome and metabolome test materials, which were correspondingly labeled as L1, L2, and L3. After quickly removing the peel and seed coat, embryos were immediately frozen in liquid nitrogen and transferred to a −80 °C freezer for storage. The remaining fruits, which were used for the analysis of physiological indices, were placed in an oven and dried at 40 °C until their weight stabilized. Meanwhile, the fruit morphology was recorded using a camera.

### 4.2. Determination of Physiological Indices during Fruit Development

From the walnut embryos, the modified Soxhlet extraction method was used to determine their oil content [50]; the anthrone-sulfuric acid method was used to determine their soluble sugar and starch contents [51]; the Kjeldahl method was used to determine their protein content [52].

### 4.3. RNA Extraction and Library Construction

Total RNA was isolated and purified using TRIzol reagent (Invitrogen, Carlsbad, CA, USA). The amount and purity of the RNA isolates from each sample were quantified using the NanoDrop ND-1000 (NanoDrop, Wilmington, DE, USA). The RNA integrity was assessed by the Bioanalyzer 2100 (Agilent, CA, USA), and was then confirmed by electrophoresis with denaturing agarose gel. After total RNA was extracted, mRNA was purified from total RNA using Dynabeads Oligo (dT). Then the cleaved RNA fragments were reverse-transcribed to create the cDNA by SuperScript™ II Reverse Transcriptase (Invitrogen, cat. 1896649, USA), which were next used to synthesize U-labeled second-stranded DNAs. Finally, a total of 9 libraries were constructed from three stages (each stage had three biological replicates). The average insert size for the final cDNA libraries was 300 ± 50 bp. We performed 2 × 150 bp paired-end sequencing (PE150) on an Illumina Novaseq™ 6000 (LC-Bio Technology Co., Ltd., Hangzhou, China).

### 4.4. Bioinformatic Analysis of RNA-Seq Data

The fastp tool (https://github.com/OpenGene/fastp, accessed on 5 January 2021) was used to remove the reads that contained adaptor contamination, low-quality bases, and undetermined bases with default parameters. The sequence quality was verified using FastQC (http://www.bioinformatics.babraham.ac.uk/projects/fastqc/, 0.11.9, accessed on 5 January 2021) [53]. Including the Q20, Q30, and GC content of the clean data. We used HISAT2 [54] to map the reads to the Walnut genome (https://www.ncbi.nlm.nih.gov/assembly/GCA_001411555.2, accessed on 6 January 2021) [24]. The mapped reads of each sample were assembled using StringTie using the default parameters.

The differentially expressed mRNAs with a fold change >2 or <0.5 were selected. The parametric F-test comparing nested linear models (*p*-value < 0.05) was employed using the R package edgeR (https://bioconductor.org/packages/release/bioc/html/edgeR.html, accessed on 6 January 2021). Genes were analyzed using the OmicStudio tool (https://www.omicstudio.cn/tool). Genes were then subjected to enrichment analysis of GO functions (http://geneontology.org, accessed on 8 January 2021) and KEGG pathways (http://www.genome.jp/kegg/, accessed on 8 January 2021). Transcription factors were predicted using PlantTFDB (http://planttfdb.gao-lab.org/, accessed on 8 January 2021) [55].

### 4.5. Quantitative Analysis

Approximately 1 μg of the total RNA obtained from each sample was used to synthesize cDNA using HiFiScript gDNA Removal RT MasterMix for qPCR (CWBIO, Beijing, China). The qPCRs were performed using MagicSYBR Mixture (CWBIO, Beijing, China). Appendix A shows the gene-specific primers for 12 genes related to lipid biosynthesis and internal control primers; the walnut GAPDH gene was used as the reference gene. The relative gene expression levels were calculated by the 2^−ΔΔCT^ method [56]. Three biological replicates were performed.

### 4.6. Metabolite Extraction

The collected samples were thawed on ice and 100 mg of powder was weighed from each sample using 1 mL of precooled 50% methanol. Then, the mixture of metabolites was vortexed for 1 min and incubated for 10 min at room temperature before being stored at −20 °C overnight. The next day, the mixture was centrifugated at 4000× *g* for 20 min. The samples were stored at −80 °C prior to the LC-MS analysis. Pooled quality control (QC) samples were also prepared by combining 10 μL of each extraction mixture [57]. Six biological replicates were executed for the extraction and subsequent analysis process.

### 4.7. LC-MS Analysis 

All samples were analyzed using a TripleTOF 5600 Plus high-resolution tandem mass spectrometer (SCIEX, Warrington, UK) with both positive and negative ion modes. Chromatographic separation was performed using an ultra-performance liquid chromatography (UPLC) system (SCIEX, UK). An ACQUITY UPLC T3 column (100 mm × 2.1 mm, 1.8 µm; Waters, UK) was used for the reverse-phase separation. For the separation of metabolites, the mobile phase consisted of solvent A (water containing 0.1% formic acid) and solvent B (acetonitrile containing 0.1% formic acid). The gradient elution conditions were as follows: a flow rate of 0.4 mL/min: 5% solvent B for 0–0.5 min; 5–100% solvent B for 0.5–7 min; 100% solvent B for 7–8 min; 100–5% solvent B for 8–8.1 min; and 5% solvent B for 8.1–10 min. The column temperature was maintained at 35 °C. 

The TripleTOF 5600 Plus system was used to detect metabolites eluted from the column. The curtain gas pressure was set at 30 PSI, while the ion source gas1 and gas2 pressure were set at 60 PSI. The interface heater temperature was 650 °C. For the positive-ion mode, the ion spray floating voltage was set at 5 kV; for the negative-ion mode, −4.5 kV. The MS data were acquired in the IDA mode. The TOF mass range was 60–1200 Da. Survey scans were acquired every 150 ms, and as many as 12 product ion scans were collected if the threshold of 100 counts/s was exceeded with a 1 + charge state. The total cycle time was fixed at 0.56 s. Four time bins were summed for each scan at a pulse frequency of 11 kHz by monitoring the 40 GHz multichannel TDC detector with four-anode/channel detection. Dynamic exclusion was set for 4 s. During the entire acquisition period, the mass accuracy was calibrated once every 20 samples. Furthermore, a QC sample was analyzed every 10 samples to evaluate the stability of the LC-MS.

### 4.8. Metabolomics Data Processing

The acquired LC-MS data pretreatment was performed using XCMS software. Each ion was identified using comprehensive retention time and *m*/*z* data. The intensity of each peak was recorded and a three-dimensional matrix containing arbitrarily assigned peak indices (retention time-*m*/*z* pairs), sample names (observations), and ion intensity information (variables) was generated. The open-access databases KEGG and HMDB were used to annotate the metabolites by matching the exact molecular mass data (*m*/*z*) to those from the database within a threshold of 10 ppm. The peak intensity data were further preprocessed using MetaX [58]. Those features that were detected in less than 50% of QC samples or 80% of biological samples were removed, the remaining peaks with missing values were imputed with the k-nearest neighbor algorithm and normalized using probabilistic quotient normalization method. In addition, the relative standard deviations of the metabolic features were calculated across all QC samples, and those with standard deviations >50% were removed.

The group datasets were normalized before analysis was performed. Data normalization was performed on all samples using the probabilistic quotient normalization algorithm. Then, QC-robust spline batch correction was performed using the QC samples. The *p*-value was analyzed by the student *t*-test, which was then adjusted for multiple tests using the FDR (Benjamini–Hochberg) for the selection of differential metabolites. The filtering conditions for the significantly altered metabolites were as follows: ratio ≥ 2 or ratio ≤ 1/2, q value ≤ 0.05, VIP ≥ 1. The metabolites were analyzed using OmicStudio tools (https://www.omicstudio.cn/tool, accessed on 30 November 2022).

### 4.9. Statistical Analyses

All statistical analyses were conducted using Statistical Package for the Social Sciences (SPSS Version 21.0 for Windows). Significant differences were determined by ANOVA and Duncan’s multiple range test (*p* < 0.05). All the data presented in this study were calculated from three independent biological replicates. Excel 2010 (Microsoft, United States) was applied to draw the bar graphs and broken line graphs. Each experiment was repeated either three or six times.

## 5. Conclusions

In conclusion, this study demonstrated that the oil and protein contents increased gradually during fruit development. Conversely, the contents of soluble sugar and starch decreased gradually during fruit development. The walnut oils reached the maximum value of 69.61% in mature embryos. We found that 50–90 DAP was the most critical period for oil synthesis. Combined metabolome and transcriptome is an effective analytical method for explaining the relationship between key genes and metabolites involved in biosynthesis pathways. The expression patterns of *LEC1*, *FUS3*, *ABI3*, and *WRI1* were consistent with the pattern of walnut oil accumulation. These TFs may play a critical role in walnut oil synthesis. *ACCase*, *KAS*, *LACS*, *FATA*, *FATB*, and *PDAT* genes may be crucial genes required for walnut oil synthesis. *FAD2* and *SAD* genes showed higher expression levels during walnut oil accumulation. Then, the contents of metabolites such as 18:1-PC, 18:2-PC, and 18:3-PC increased significantly with progressing fruit development. It is noteworthy that high expression levels of *FAD2* and *SAD* genes may be conducive to the rapid accumulation of walnut unsaturated fatty acid, as they are key genes for lipid synthesis. The walnut lipid profile and the lipid metabolism pathway constructed here have important theoretical and practical value for further study of walnut lipid metabolism and functional development. In addition, further studies on their functions in FA biosynthesis and oil accumulation are required.

## Figures and Tables

**Figure 1 plants-12-00538-f001:**
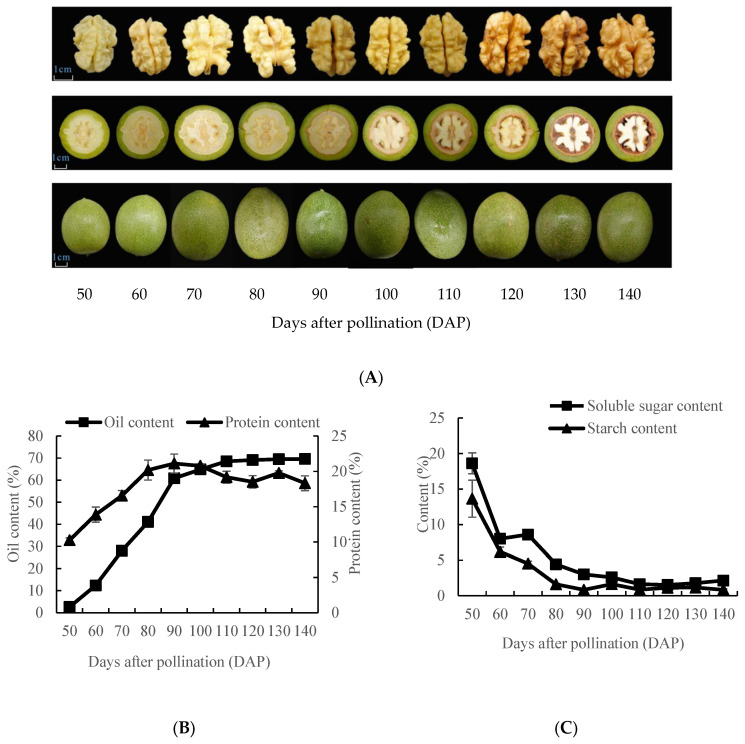
Changes in the morphological characteristics and indices of walnut fruits during development. (**A**) Fruit morphological characteristics at 50–140 DAP. The scale bar is 1 cm. The walnut embryos, transverse cuts of walnut fruits, and appearance of walnut fruit are shown in the above, middle, and below panels. (**B**) Embryo oil and protein content at 50–140 DAP. The *y*-axis on the left shows the oil content; *y*-axis on the right shows the protein content; the *x*-axis shows the different developmental stages. (**C**) Embryo soluble sugar and starch content at 50–140 DAP. The *y*-axis shows the soluble sugar and starch content; the *x*-axis shows the different developmental stages. The data were analyzed using three biological replicates. Standard deviations are shown with error bars. DAP, days after pollination.

**Figure 2 plants-12-00538-f002:**
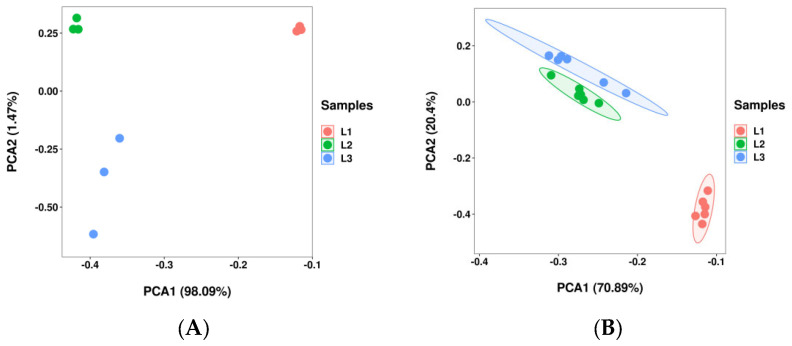
Principal component analyses (PCAs) of transcripts and metabolites during walnut fruit development. Note: (**A**) PCA score of transcriptomic data and (**B**) PCA score of metabolomic data. The *x*-axis represents principal component 1 (PC1); the *y*-axis represents principal component 2 (PC2). The three time points are distinguished by different colors. L1 was 60, L2 was 90, and L3 was 120 DAP.

**Figure 3 plants-12-00538-f003:**
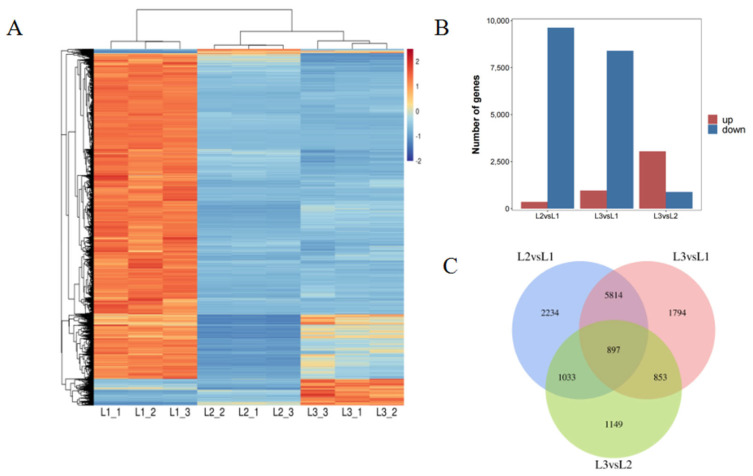
Characteristics and expression analysis of DEGs in walnut embryos at different developmental stages. (**A**) Heat map of DEGs based on FPKM. The color scale represents the calculated Z-score. Red and blue indicate higher and lower abundances, respectively. (**B**) Number of DEGs in the different combinations. The blue bar denotes downregulated genes; the red bar denotes upregulated genes. (**C**) Venn diagram of DEGs in the different combinations. L1 was 60, L2 was 90, and L3 was 120 DAP. (L2−VS−L1, L3−VS−L1, L3−VS−L2; “a” was the experimental group; “b” was the control in “a−vs−b”).

**Figure 4 plants-12-00538-f004:**
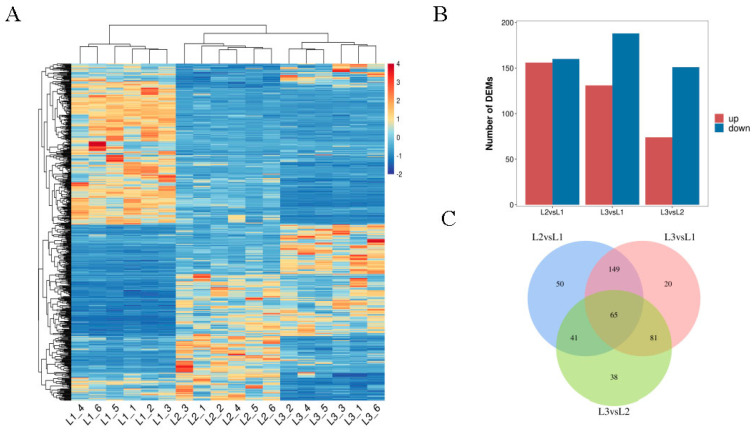
Characteristics and analysis of DEMs in walnut embryos at different developmental stages. (**A**) Heat map of DEMs based on the peak area. The color scale represents the calculated Z-score. The red blocks indicate upregulated metabolites; the blue blocks indicate downregulated metabolites. (**B**) Number of DEMs in different combinations. The blue bar denotes downregulated metabolites; the red bar denotes upregulated metabolites. (**C**) Venn diagram of DEMs in different combinations. L1 was 60 DAP, L2 was 90 DAP, and L3 was 120 DAP. (L2−VS−L1, L3−VS−L1, L3−VS−L2; “a” was the experimental group; “b” was the control in “a−vs−b”).

**Figure 5 plants-12-00538-f005:**
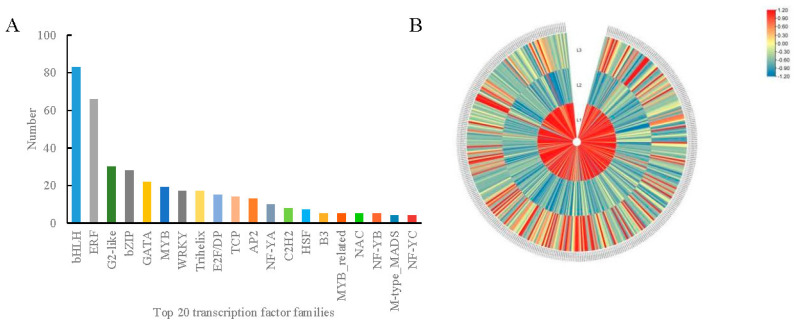
Analysis of differentially expressed transcription factors in walnut embryos at different developmental stages. (**A**) Differentially expressed transcription factor families (top 20); (**B**) cluster analysis of differentially expressed transcription factors based on FPKM. L1 was 60, L2 was 90, and L3 was 120 DAP. The color scale represents the calculated Z-score. Red and blue indicate higher and lower abundances, respectively.

**Figure 6 plants-12-00538-f006:**
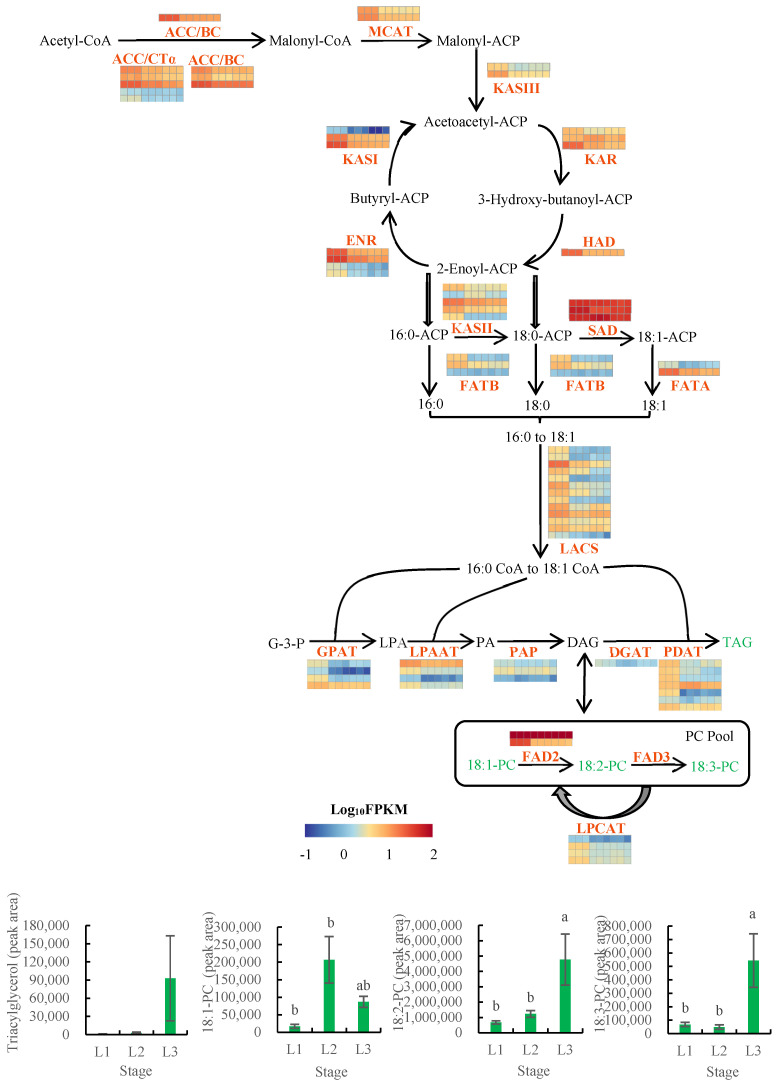
Analysis of lipid biosynthesis genes and metabolites in walnut embryos. Note: the expression values of genes are shown in the thermal map, which displays the homogenization values of genes. Red and blue indicate higher and lower abundances, respectively. L1 was 60 DAP, L2 was 90 DAP, and L3 was 120 DAP. The nine squares in each horizontal row correspond to the nine samples (L1_1, L1_2, L1_3, L2_1, L2_2, L2_3, L3_1, L3_2, and L3_3). The metabolite peak areas are shown in the cylindrical map. Bars represent the mean ± SE (n = 6). Bars with different lowercase letters (a and b) are significantly different (*p* < 0.05). ACC/CTα, acetyl−CoA carboxyl transferase subunit alpha; ACC/BC, acetyl−CoA carboxylase biotin carboxylase subunit; ACC/BCCP, acetyl−CoA carboxylase biotin carboxyl carrier protein; MCAT, malonyl CoA−acyl carrier protein transacylase; KASI, 3−oxoacyl−ACP synthase I; KASII, 3−oxoacyl−ACP synthase II; KASIII, 3−oxoacyl−ACP synthase III; KAR, 3−oxoacyl−ACP reductase; HAD, 3−hydroxyacyl−ACP dehydratase; ENR, enoyl−ACP reductase; SAD, stearoyl−ACP-desaturase; FATA, fatty acyl−ACP thioesterase A; FATB, fatty acyl−ACP thioesterase B; LACS, long−chain acyl−CoA synthetase; GPAT, glycerol−3−phosphate acyltransferase; LPAAT, lysophosphatidic acid acyltransferase; PAP, phosphatidate phosphatase; PDAT, phospholipid:diacylglycerol acyltransferase; DGAT1, diacylglycerol acyltransferase 1; FAD2, fatty acid desaturase 2; FAD3, fatty acid desaturase 3; LPCAT, lysophospholipid acyltransferase; G−3−P, Glycerol−3−phosphate; LPA, 1−acylglycerol−3P; PA, 1,2−diacylglycerol−3P; DAG, 1,2−diacylglycerol; TAG, triacylglycerol.

**Figure 7 plants-12-00538-f007:**
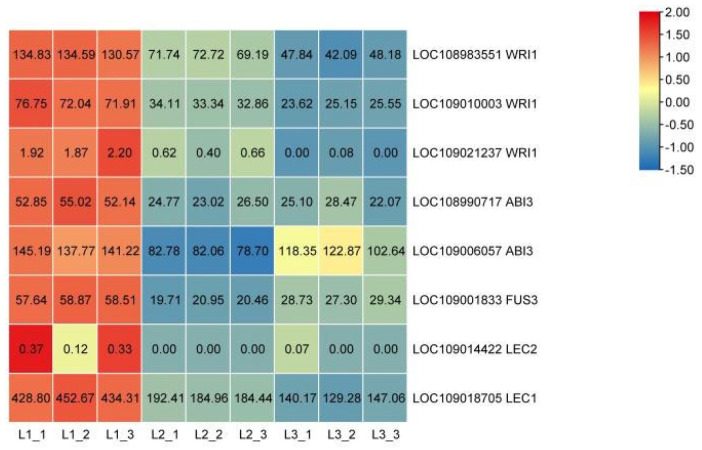
Expression heat map of lipid biosynthesis differential transcription factors based on FPKM. The color scale represents the calculated Z-score. Red and blue indicate higher and lower abundances, respectively. L1 was 60 DAP, L2 was 90 DAP, and L3 was 120 DAP. WRI1, ethylene-responsive transcription factor WRI1; ABI3, B3 domain-containing transcription factor ABI3; FUS3, B3 domain-containing transcription factor FUS3; LEC1, nuclear transcription factor Y subunit B-6; LEC2, B3 domain-containing transcription factor LEC2.

**Figure 8 plants-12-00538-f008:**
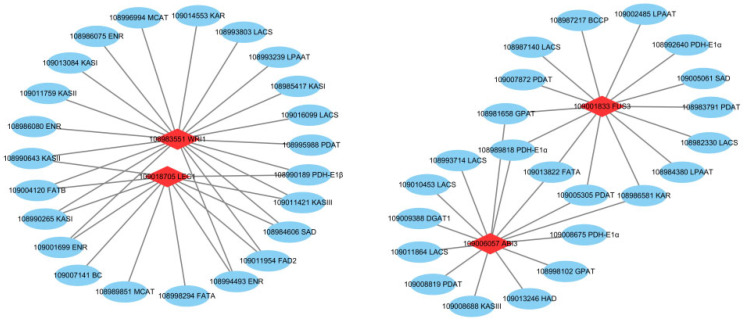
Connection network between lipid synthesis-related transcription factors (red rhombus) and structural genes (blue circle). The correlation coefficients were greater than 0.9, both with the TFs and structural genes. The gray line indicates a positive correlation.

**Figure 9 plants-12-00538-f009:**
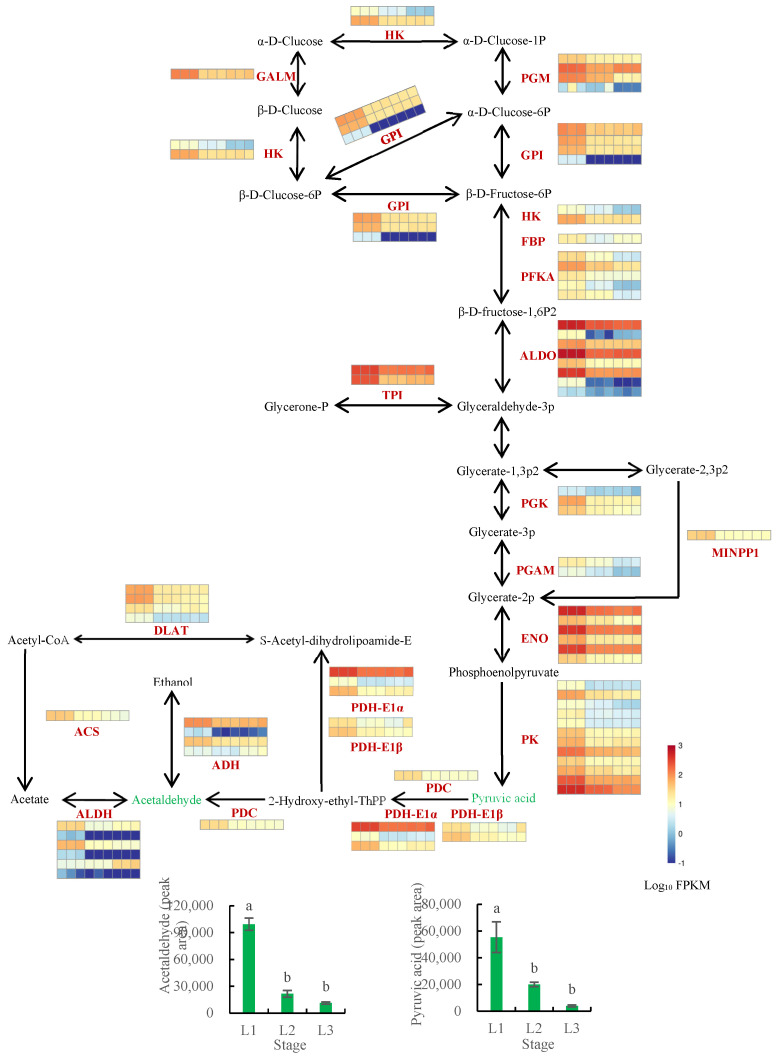
Analysis of the glycolysis biosynthesis pathway in walnut embryos. Note: the metabolite peak areas are shown in the cylindrical map, while the expression values of genes are in the thermal map, which shows the homogenization values of genes. The nine squares in each horizontal row correspond to the nine samples (L1_1, L1_2, L1_3, L2_1, L2_2, L2_3, L3_1, L3_2, and L3_3). L1 was 60 DAP, L2 was 90 DAP, and L3 was 120 DAP. Bars represent mean ± SE (n = 6). Bars with different lowercase letters (a and b) are significantly different (*p* < 0.05). HK, hexokinase; PGM, phosphoglucomutase; GALM, aldose 1−epimerase; GPI, glucose−6−phosphate isomerase; FBP, fructose−1,6−bisphosphatase; PCKA, phosphoenolpyruvate carboxykinase (ATP); ALDO, fructose-bisphosphate aldolase; TPI, triosephosphate isomerase; PGK, phosphoglycerate kinase; PGAM, 2,3−bisphosphoglycerate−dependent phosphoglycerate mutase; ENO, enolase; PK, pyruvate kinase; MINPP1, multiple inositol polyphosphate phosphatase1; PDC, pyruvate decarboxylase; PDH−E1α, pyruvate dehydrogenase E1 component alpha subunit; PDH−E1β, pyruvate dehydrogenase E1 component beta subunit; ADH, alcohol dehydrogenase; ALDH, aldehyde dehydrogenase; ACS, acetyl−CoA synthetase; DLAT, pyruvate dehydrogenase E2 component.

**Figure 10 plants-12-00538-f010:**
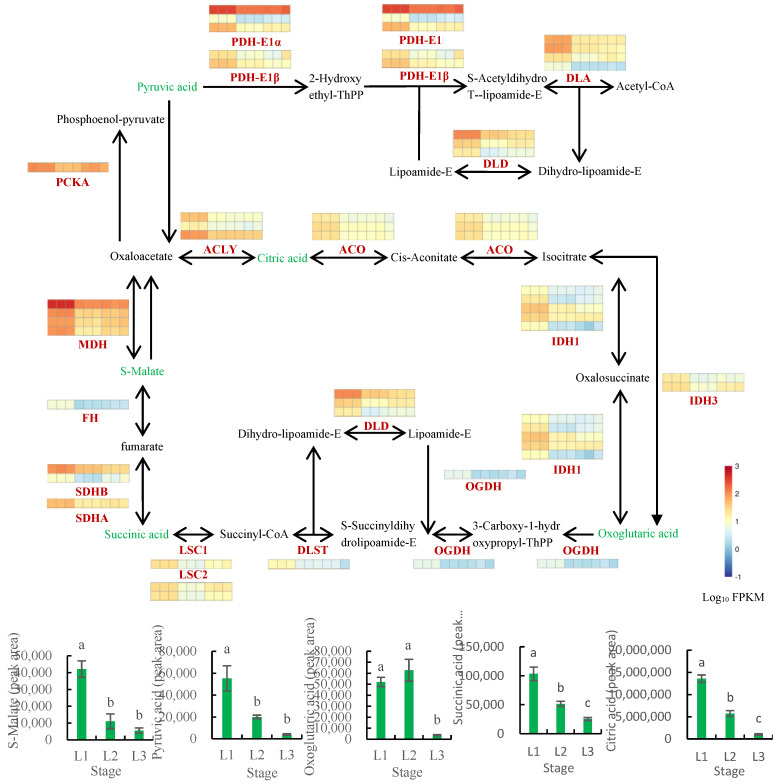
Analysis of the TCA cycle pathway in walnut embryos. Note: the metabolite peak areas are shown in the cylindrical map, while the expression values of genes are in the thermal map, which shows the homogenization values of genes. L1 was 60 DAP, L2 was 90 DAP, and L3 was 120 DAP. The nine squares in each horizontal row correspond to the nine samples (L1_1, L1_2, L1_3, L2_1, L2_2, L2_3, L3_1, L3_2, and L3_3). Bars represent the mean ± SE (n = 6). Bars with different lowercase letters (a, b, c) are significantly different (*p* < 0.05). PDH−E1α, pyruvate dehydrogenase E1 component alpha subunit; PDH−E1β, pyruvate dehydrogenase E1 component beta subunit; DLAT, pyruvate dehydrogenase E2 component; DLD, dihydrolipoamide dehydrogenase; PCKA, phosphoenolpyruvate carboxykinase (ATP); ACLY, ATP−citrate synthase; ACO, aconitate hydratase; IDH1, isocitrate dehydrogenase (NADP); IDH3, isocitrate dehydrogenase (NAD+); OGDH, 2−oxoglutarate dehydrogenase; DLST, dihydrolipoyl lysine-residue succinyl transferase; LSC1, succinyl−CoA synthetase alpha subunit; LSC2, succinyl−CoA synthetase beta subunit; SDHB, succinate dehydrogenase (ubiquinone) iron−sulfur subunit; SDHA, succinate dehydrogenase (ubiquinone) flavoprotein subunit; FH, fumarate hydratase; MDH, malate dehydrogenase.

**Figure 11 plants-12-00538-f011:**
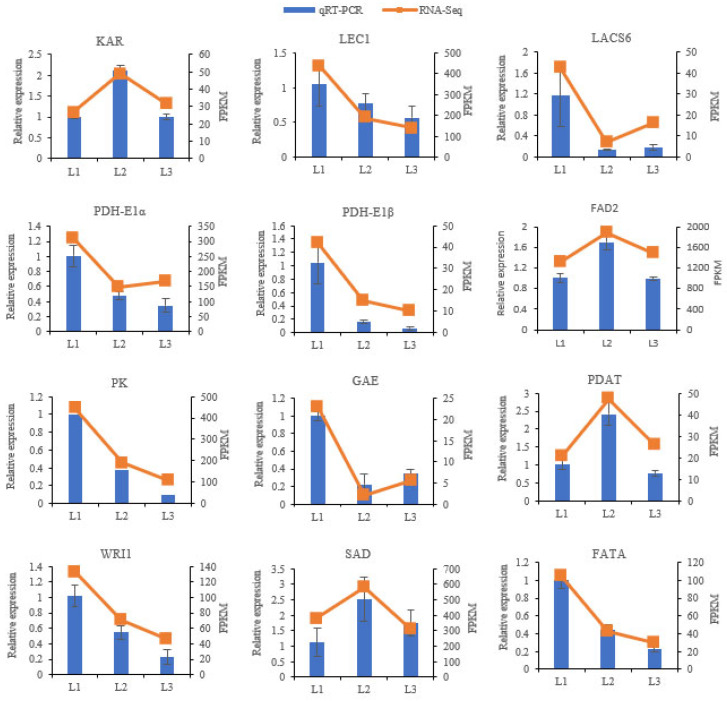
Expression patterns of twelve genes, as determined by the transcriptomic data and qRT−PCR results. The *y*-axis on the right represents the FPKM value obtained by RNA−seq; the *y*-axis on the left shows the relative gene expression levels (2^−ΔΔ^*^C^*^t^) analyzed by qRT−PCR. The *x*-axis represents the different samples. Bars represent mean ± SE (n = 3). FPKM represents the fragments per kilobase per million. L1 was 60 DAP, L2 was 90 DAP, and L3 was 120 DAP. KAR, 3−oxoacyl−ACP reductase; LEC1, nuclear transcription factor Y subunit B−6; LACS6, long−chain acyl−CoA synthetase 6; PDH−E1α, pyruvate dehydrogenase E1 component alpha subunit; PDH−E1β, pyruvate dehydrogenase E1 component beta subunit; FAD2, fatty acid desaturase 2; PK, pyruvate kinase; GAE, UDP−glucuronate 4−epimerase; PDAT, phospholipid:diacylglycerol acyltransferase; WRI1, ethylene−responsive transcription factor WRI1; SAD, stearoyl−ACP−desaturase; FATA, fatty acyl−ACP thioesterase A.

## Data Availability

The data presented in this study are available in Appendix A.

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
