# Peer review of "Metabolomics and Transcriptomics Provide Insights into Lipid Biosynthesis in the Embryos of Walnut (Juglans regia L.)"

_plants, 2023, doi:10.3390/plants12030538_

Round 1

Reviewer 1 Report

This is a useful study of development of walnut fruit, focusing on the oils. The authors should provide more background to what is already known about walnut fruit and oil development, and also justification for the analytical strategy they adopted. There are also some details that need to be added. I make suggestions below.

In the Introduction, lines 42 - 59, if it is necessary to describe the pathway for biosynthesis of fatty acids and triacylglycerol, a diagram would be much better than a written description. The text can then state any interesting or unusual features that happen in walnut.

Is anything already known about the chain length and degree of unsaturation typical of TAG in walnut oil and any changes as the fruit develops? If so, please add to introduction, or say it is unknown.

Has the development of walnut fruit already been studied? If so, a diagram and short outline in the Introduction would be useful to set the scene.

Is there already genomic data and resources for the walnut Juglans regia? Might be worth a few sentences explaining the current situation.

The introduction also needs something about the strategy taken to obtain genomic and metabolite data - why particular technique and data analysis methods used.

Before section 2.2, that describes analysis of RNA-seq data, there needs to be a section about the experimental design and processing to obtain the data. There needs to be a description of where samples L1, L2 and L3 came from. This information is also important in order to evaluate section 2.3. Lines 159 - 161 are about the time period when the samples were collected. This needs to be placed much earlier, along with the nature of the replicates (biological, technical, all walnuts from one tree ... What variety of walnut ... ). Sections 4.1 and 4.3 do not describe the samples and experimental design fully. 

Section 2.4 also needs to have information about the origin and analysis strategy of the metabolites placed before it. 

Author Response

Dear Ms. Supakorn Nundaeng and Reviewers

Thank you very much for arranging a timely review for our manuscript and the hard work of the Associate Editor/Reviewers. We have carefully evaluated the Editor/Reviewers’ critical comments and thoughtful suggestions, responded to these suggestions as follow as Q&A format point-to-point, and revised the manuscript accordingly. All changes made to the text are indicated in our revised manuscript by highlighting so that you may be easily identified. With regard to the Associate Editor/Reviewers’ comments and suggestions, we wish to reply as follows:

Authors' Responses to Reviewer's Comments (Reviewer 1)

Q1: 1. In the Introduction, lines 42 - 59, if it is necessary to describe the pathway for biosynthesis of fatty acids and triacylglycerol, a diagram would be much better than a written description. The text can then state any interesting or unusual features that happen in walnut.

A1: Thank you very much for your suggestion. We have drawn a diagram involved oil synthesis (Figure 6). We have refined this paragraph in the revised version (Line 52-54).

Q2: 2. Is anything already known about the chain length and degree of unsaturation typical of TAG in walnut oil and any changes as the fruit develops? If so, please add to introduction, or say it is unknown.

A2: According to your suggestion, we have provided more information about unsaturation typical of TAG and changes as the fruit develops in the revised version (Line 41-45). Thank you very much for your correction.

Q3: 3. Is there already genomic data and resources for the walnut Juglans regia? Might be worth a few sentences explaining the current situation.

A3: We are sorry for our negligence. We have added the information of walnut genome resources in the revised version (Line 71-73).

Q4: 4. The introduction also needs something about the strategy taken to obtain genomic and metabolite data - why particular technique and data analysis methods used.

A4: We are sorry for our negligence. We have added relevant information to explain why transcriptomics and metabolomics were used in the revised version (Line 73-81).

Q5: 5. Before section 2.2, that describes analysis of RNA-seq data, there needs to be a section about the experimental design and processing to obtain the data. There needs to be a description of where samples L1, L2 and L3 came from. This information is also important in order to evaluate section 2.3. Lines 159 - 161 are about the time period when the samples were collected. This needs to be placed much earlier, along with the nature of the replicates (biological, technical, all walnuts from one tree ... What variety of walnut ... ). Sections 4.1 and 4.3 do not describe the samples and experimental design fully. 

A5: We are sorry for our negligence. We have added describes analysis of RNA-seq data (the experimental design and processing to obtain the data) in the revised version (Line 136-139). Line 159 - 161 have been placed at the top of the paragraph in the revised version (Line 169-173). We have fully described section 4.1 and 4.3 in the revised version (Line 642-651 and Line 661-666). We hope that our modifications can meet your requirements. Thank you very much for your suggestion.

Q6: 6. Section 2.4 also needs to have information about the origin and analysis strategy of the metabolites placed before it. 

A6: We are sorry for our negligence. We have added those information in the revised version (Line 200-202). Thank you very much for your suggestion.

Thanks again for your and Reviewers’ comments and suggestions. We hope that the revised version can meet your requirements. We look forward to hearing from you soon.

Sincerely yours,

Guohui Qi

College of Forestry, Hebei Agricultural University, Baoding 071001, China

E-mail: bdqgh@sina.com

Minsheng Yang

College of Forestry, Hebei Agricultural University, Baoding 071001, China

E-mail: yangms100@126.com

Reviewer 2 Report

Manuscript title:  Metabolomics and Transcriptomics  Provide Insights into  Lipid Biosynthesis in the Embryos of Walnut (Juglans regia L.)

This study has certain significance in the metabolomics and transcriptomics of walnut embryos.

 However, revisions are necessary for the current version of the manuscript. The following questions to be addressed/considered may be helpful to improve the manuscript.

Major comments

·       Insufficient Abstract: In the abstract, the main aim and background of the manuscript are missing, the current version it only highlights the result. In addition, it would be even better to have a sentence as a future perspective.

·       The unit/abbreviation is not mentioned before, consider defining the abbreviation when mentioned for the first time…. Please check throughout the manuscript to define the abbreviations.

·       Line 71-76, the aim or hypothesis of the study is clear, however, the approach is missing ….

·       Lake of scientific literature to support the statements and findings throughout the manuscript…... I have made some suggestions for that and more need it….

·       More information is needed for ALL TABLE captions and define the abbreviation and units that are used. And adjust the significant figures for the table and manuscript.

·       I have a major concern about the results and discussion section. The authors describe the results and compare the results with previous studies, however, insight mechanisms are still insufficient.

Specific comments:

Abstract

If the unit/abbreviation is not mentioned before, consider defining the abbreviation when mentioned for the first time.

Introduction:

Line 37-41: A complicated sentence, please revise and check the grammar

Line 35: A reference is needed here, for example, you can use: https://doi.org/10.1007/s12161-021-02203-0

Line 46: A reference is needed here, for example, you can use: https://doi.org/10.1016/j.bbalip.2015.12.002

Line 49-57: A reference is needed here…….

Line 71-76: A complicated sentence, please revise and check the grammar

R&D section

Figure 1: the labeling for the first panel is missing, I guess it is ‘’A’’. And what fruit morphological characteristics represent with the 3-row figures?

For ‘’B’’ indicate which axis represents which!

For Figure 2, 3, 4 5, 7, 8 . The figure are blur, even through I have printed the file – the quality of the figure can be improved. And the text is hard to read, maybe because it too small.

Line 148-270: Instead of describing the data, it is better to discuss what the data means, and how we understand the data.

These sections are repeating information already presented and explain things in an unnecessarily complicated way. The quality of the manuscript would benefit from the whole section being condensed, Line 241-22, Line 394-411, Line 482-499.

In MM section

Literature references are missing for all sub-section. It would be better to cite the references that the procedure adopted.

Additional info is needed for the table caption, most importantly significant figures.

In MM section, what is the quality control (QC) data? There is no mention of the QC.

What is the accuracy of the instruments, recovery, LOD, and LOQ ……. These parameters are needed to report the efficiency of any analytical system.

In general, how many times you’ve recorded the data,? duplicate? Triplicate?..... what you mentioned in the text is not clear, please elaborate more on this

Conclusion

I believe there are other important conclusions that could be made from this study…. And the future perspectives for the following research are highly crucial here.

Author Response

Dear Ms. Supakorn Nundaeng and Reviewers

Thank you very much for arranging a timely review for our manuscript and the hard work of the Associate Editor/Reviewers. We have carefully evaluated the Editor/Reviewers’ critical comments and thoughtful suggestions, responded to these suggestions as follow as Q&A format point-to-point, and revised the manuscript accordingly. All changes made to the text are indicated in our revised manuscript by highlighting so that you may be easily identified. With regard to the Associate Editor/Reviewers’ comments and suggestions, we wish to reply as follows:

Authors' Responses to Reviewer's Comments (Reviewer 2)

Q1: 1. Insufficient Abstract: In the abstract, the main aim and background of the manuscript are missing, the current version it only highlights the result. In addition, it would be even better to have a sentence as a future perspective.

A1: Thanks, we are sorry for our negligence. We have added the main aim and background in the revised version (Line 7-9). We also add a future perspective in the revised version (Line 29-30). I hope that our modifications can meet your requirements. Based on these results, we will further study the regulatory mechanism of oil synthesis in future. Thank you very much for your suggestion.

Q2: 2. The unit/abbreviation is not mentioned before, consider defining the abbreviation when mentioned for the first time…. Please check throughout the manuscript to define the abbreviations.

A2: Thank you very much for your suggestion. We are sorry for our negligence. We have modified the manuscript to define the unit/abbreviation.

Q3: 3. Line 71-76, the aim or hypothesis of the study is clear, however, the approach is missing ….

A3: Thank you very much for your suggestion. We are very sorry for this mistake. We have modified this sentence according to the comment in the revised version (Line 82-87).

Q4: 4. Lake of scientific literature to support the statements and findings throughout the manuscript…... I have made some suggestions for that and more need it….

A4: Thank you very much for your suggestion. We have added relevant literature to supports the statements and findings throughout the manuscript.

Q5: 5. More information is needed for ALL TABLE captions and define the abbreviation and units that are used. And adjust the significant figures for the table and manuscript.

A5: We are sorry for our negligence. Thanks, we have supplemented the information (Table S3-6) for the figure 5A, 6, 9, and 10. We have modified the use of abbreviations and units in the legend of figures.

Q6: 6. I have a major concern about the results and discussion section. The authors describe the results and compare the results with previous studies, however, insight mechanisms are still insufficient.

A6: We are very pleased that the Reviewer is interested in this result. We are very sorry that we did not meet the requirement of the Reviewer. We will try our best to explore the mechanism of oil synthesis. Thank you very much for this precious suggestion.

Specific comments:

Abstract

Q1: 1. If the unit/abbreviation is not mentioned before, consider defining the abbreviation when mentioned for the first time.

A1: Sorry for this mistake. We have carefully examined the manuscript and modified the mistake. The Reviewer is very precise and careful. Thank you very much for your correction.

Introduction:

Q2: 2. Line 37-41: A complicated sentence, please revise and check the grammar

A2: Thanks, we have revised the sentence in the revised version (Line 39-40).

Q3: 3. Line 35: A reference is needed here, for example, you can use: https://doi.org/10.1007/s12161-021-02203-0

A3: Thanks, we have added this reference in the revised version (Line 36).

Q4: 4. Line 46: A reference is needed here, for example, you can use: https://doi.org/10.1016/j.bbalip.2015.12.002

A4: Thanks, we have added this reference in the revised version (Line 49).

Q5: 5. Line 49-57: A reference is needed here…….

A5: Thanks, we have added a referencein in the revised version (Line 54).

Q6: 6. Line 71-76: A complicated sentence, please revise and check the grammar

A6: Thanks, we have revised the sentence in the revised version (Line 82-87).

R&D section

Q7: 7. Figure 1: the labeling for the first panel is missing, I guess it is ‘’A’’. And what fruit morphological characteristics represent with the 3-row figures? For ‘’B’’ indicate which axis represents which

A7: We are sorry for our negligence. The legend of figure 1 has been modified according to your suggestions in the revised version (Line 130-133). Thank you very much for your correction.

Q8: 8. For Figure 2, 3, 4 5, 7, 8 . The figure are blur, even through I have printed the file – the quality of the figure can be improved. And the text is hard to read, maybe because it too small.

A8: Thank you very much for your correction. Figure 2, 3, 4, 5, 7, 8 have been improved according to your suggestions (we have provided the original images and put it in the compressed file). The Reviewer is very precise and careful. Thank you very much for your suggestion.

Q9: 9. Instead of describing the data, it is better to discuss what the data means, and how we understand the data.

A9: We are very pleased that the Reviewer is interested in this result. We will try our best to explain the meaning of the data in the revised version (Line 164-491). Thank you very much for this precious suggestion.

Q10: 10. These sections are repeating information already presented and explain things in an unnecessarily complicated way. The quality of the manuscript would benefit from the whole section being condensed, Line 241-22, Line 394-411, Line 482-499.

A10: We are sorry for our negligence. We have deleted the unnecessarily contents and refined the content in the revised version ( Line 260-306, Line 402-413, and Line 478-491). We hope that our modifications can meet your requirements. Thank you very much for your suggestion.

In MM section

Q11: 11. Literature references are missing for all sub-section. It would be better to cite the references that the procedure adopted.

A11: We are sorry for our negligence. We have add references at materials and methods section in the revised version ( Line 673-727).

Q12: 12. Additional info is needed for the table caption, most importantly significant figures.

A12: We are sorry for our negligence. Thanks, we have supplemented the information (Table S3-6) for the figure 5A, 6, 9, and 10. 

Q13: 13. In MM section, what is the quality control (QC) data? There is no mention of the QC.

A13: The Reviewer is very precise and careful. Thank you very much for your correction. We have provided QC data: table S1-2 (transcriptomics) in the revised version (Line 164-168 and Line 671-673) and figure S1 (untargeted metabolomics) in the revised version (Line 204-206).

Q14: 14. What is the accuracy of the instruments, recovery, LOD, and LOQ ……. These parameters are needed to report the efficiency of any analytical system.

A14: Thanks, the targeted metabolomics contains these parameters (the instruments, recovery, LOD, and LOQ), but the untargeted metabolomics does not contain these parameters. Untargeted Metabolomics (LC-MS) were used in this study. We have provided figure S1 (Total ion chromatogram) in the revised version (Line 202-204). The overlaps of QC samples in TIC can be used to preliminarily judge the state of the instrument. The higher intensity overlap presents, the more stable the instrument is. In addition, PCA analysis of metabolomics (Figure 2B) can also reflect the accuracy of data. Our description of metabolome experimental method is consistent with that in literature (Age-related changes in metabolites in young donor livers and old recipient sera after liver transplantation from young to old rats[J]. Aging Cell, 2021).

Q15: 15. In general, how many times you’ve recorded the data,? duplicate? Triplicate?..... what you mentioned in the text is not clear, please elaborate more on this

A15: We are sorry for our negligence. We have added these information: transcriptomics (three biological replicates); untargeted metabolomics (six biological replicates); physiological indices (three biological replicates) in the revised version (Line 642-644).

 Conclusion

Q16: 16. I believe there are other important conclusions that could be made from this study…. And the future perspectives for the following research are highly crucial here.

A16: Thank you for your suggestion. We have added other important conclusions and the future perspectives in the revised version (Line 750-763).

Thanks again for your and Reviewers’ comments and suggestions. We hope that the revised version can meet your requirements. We look forward to hearing from you soon.

Sincerely yours,

Guohui Qi

College of Forestry, Hebei Agricultural University, Baoding 071001, China

E-mail: bdqgh@sina.com

Round 2

Reviewer 2 Report

The revised manuscript has improved compared to the original version. The authors tried to address my questions as much as possible. I recommend the manuscript to be published!